# A Controlled Clinical Trial on the Effects of Aquatic Exercise on Cognitive Functions in Community-Dwelling Older Adults

**DOI:** 10.3390/brainsci14070703

**Published:** 2024-07-13

**Authors:** Renata Terra de Oliveira, Tayla Borges Lino, Gabriella Simões Scarmagnan, Suzi Rosa Miziara Barbosa, Ana Beatriz Gomes de Souza Pegorare, Gustavo Christofoletti

**Affiliations:** School of Medicine, Institute of Health, Federal University of Mato Grosso do Sul—UFMS, Campo Grande 79060-900, Brazil; reoliveira_terra@hotmail.com (R.T.d.O.); tayla.lino@ufms.br (T.B.L.); simoes.gabriella@gmail.com (G.S.S.); suzi.barbosa@ufms.br (S.R.M.B.); ana.pegorare@ufms.br (A.B.G.d.S.P.)

**Keywords:** cognition, executive functions, hydrotherapy, aquatic therapy, controlled clinical trial, aged, cognitive aging

## Abstract

Several therapies have been developed to reduce cognitive decline associated with aging. Aquatic exercises, which are widely used to enhance functional capacity, may play a role in stimulating cognitive functions. This study investigated the effects of a 3-month aquatic exercise program on cognitive functions in community-dwelling older adults. In this prospective, single-blinded, controlled clinical trial, 31 participants were allocated to either the experimental (aquatic exercises) or control (no-exercise) group. The intervention program consisted of exercises conducted twice a week in a 1.2 m deep indoor pool. The main outcome measures were cognitive functions, assessed using Raven’s Progressive Matrices test and the Wisconsin Card Sorting Test. A repeated-measures analysis of variance was used to assess the impact of the exercise program. The effect sizes (*η*^2^*p*) were reported when a level of significance was achieved (*p* < 0.05). Compared with the control group, the participants who underwent aquatic exercises showed positive outcomes in Raven’s Progressive Matrices test (*p* = 0.046; *η*^2^*p* = 0.131) and the Wisconsin Card Sorting Test *(p* = 0.001, *η*^2^*p* = 0.589). Complementary analyses of the Wisconsin Card Sorting Test indicated that the benefits of the aquatic exercise were observed in terms of the number of trials (*p* = 0.001, *η*^2^*p* = 0.478), number of errors (*p* = 0.001, *η*^2^*p* = 0.458), and number of non-perseverative errors (*p* = 0.001, *η*^2^*p* = 0.302). The results indicate that a period of three months of aquatic exercise was beneficial for stimulating specific aspects of the cognitive function of community-dwelling older individuals. Aquatic exercise should be prescribed to this population.

## 1. Introduction

The world has undergone a demographic transition caused by a decline in birth rates and an increase in life expectancy [1]. Today, people live longer and have a better quality of life [2]. Access to healthcare services and new healthy habits are changing the appearance of older people in society. In the past, older individuals were labeled a burden because of their physical and cognitive decline. Now, older people hold prominent positions and are important in both families and work [3,4,5].

Despite living longer and having a better quality of life, the physical and cognitive decline caused by aging cannot be avoided. Individual variations linked to daily habits, socioeconomic factors, and genetics make the routine activities of older people easier or more difficult [6]. Previous studies have focused on the effects of aging on functional capacity, as it is directly related to one’s independence [6,7,8]. With increasing life expectancy and the subsequent increase in the prevalence of brain diseases, studies are now focusing on the effects of aging on cognitive functions [9,10,11].

Cognitive functions are defined as the mental processes of thinking and understanding. They involve a set of abilities that are important for perceiving, reacting, processing, making decisions, and producing appropriate responses to an environment [12,13,14]. Cognitive functions are concentrated in the prefrontal cortex [15]. However, their association with other brain areas provide important responses related to memory, attention, orientation, executive functions, praxis, gnosis, language, and visual–spatial skills [16,17,18].

Previous studies have suggested a series of cognitive and motor interventions as a way to identify slowed executive processing, predict cognitive impairment, and maintain brain functions that are known to decline with age [19,20,21]. Currently, there is no consensus for the benefits of these interventions in promoting neurogenesis in the brain [22,23]. However, some effects of these interventions are related to the production of neurotransmitters and other proteins, such as brain-derived neurotrophic factor, which are beneficial for older individuals as they are associated with cognitive reserve and executive function performance [24,25,26].

Aquatic exercise is an important treatment option for older individuals. Its potential in promoting healing and reducing pain was demonstrated as far back as ancient Egyptian and Greek medicine [27]. The effects of aquatic exercises have been linked to fluid mechanics [28]. Hydrostatic pressure, density, buoyancy, viscosity, and resistance play crucial roles in the treatment and recovery of older adults. Individuals with musculoskeletal or neurologic disorders who receive aquatic exercise may experience shorter recovery periods than those who undergo other treatments [29,30]. However, the potential benefits of aquatic exercises on cognitive functions require further investigation [31,32]. The current body of research is limited, likely due to the small number of studies addressing this topic.

One of the few studies addressing the impact of aquatic exercise on cognitive function was conducted by Häfele et al. [33]. The researchers assessed 52 participants, with 35 who underwent water-based aerobic training and 17 who remained sedentary in the control group. The findings showed no benefits of the intervention in terms of their scores on the Mini-Mental State Examination. This raises the question of whether aquatic exercise alone can enhance cognitive function in older individuals compared to a combination of water and cognitive training.

In contrast, Campbell et al. [34] used a different approach. The researchers enrolled 31 individuals in a water exercise program combined with cognitive training, in which the control group of 36 participants underwent land-based exercise with cognitive training. Notably, all participants in this study had mild cognitive impairment, unlike in the study of Häfele et al. [33]. The authors found that individuals in the aquatic-based exercise plus cognitive training group experienced significant cognitive benefits, whereas those in the land-based exercise plus cognitive training group showed limited improvement. The results suggest that aquatic exercises combined with cognitive training can improve cognition in individuals with mild cognitive impairment. However, it remains unclear whether aquatic exercises improve cognition in cognitively preserved older adults.

With this context in mind, this study aimed to investigate the potential benefits of a 3-month aquatic exercise program on the cognitive functions of community-dwelling older adults. We hypothesized that the individuals who underwent aquatic training would experience improvements in their cognitive function (assessed using Raven’s Progressive Matrices test and the Wisconsin Card Sorting Test) compared to that of a control group that did not engage in any exercise program. In this study, we believed that incorporating concurrent aquatic exercise would optimize brain plasticity and mitigate age-related decline.

## 2. Materials and Methods

This was a single-blinded, controlled clinical trial with two parallel groups. This study was conducted at the Outpatient Clinic of the Federal University of Mato Grosso do Sul, Campo Grande, Brazil. We followed the CONSORT statement checklist [35], Declaration of Helsinki, and guidelines for good clinical practice. The participants signed a consent form before the assessment. The protocol was approved by the institution’s ethics committee (# 4.350.403), and it was prospectively registered in the Brazilian Registry of Clinical Trials (# RBR-6q2wx4t).

The sample was considered convenient because only participants who were willing to undergo a medical examination prior to the pool activities were included. Recruitment involved direct contact with potential participants and the use of social media platforms to reach a wider audience. 

The sample consisted of 31 individuals, 17 women and 14 men, with a mean age of 68.1 years (SD, 5.5), who were allocated to either the experimental (intervention) or control (no intervention) group. Individuals were eligible for inclusion if they were at least 60 years old, sedentary, living in a private residence in the community, of any creed or race, and did not present any walking problems. Exclusion criteria consisted of participants with a previous history of neurological or psychiatric disorders as well as participants who were deemed medically unfit for engaging in pool activities. All the participants were screened for dementia. Individuals who exhibited scores on the Mini-Mental State Examination test that were lower than the cutoff values adjusted for education and age established for the local population were excluded from the study [36,37].

### 2.1. Sample Size, Blinding, and Randomization

The sample size was determined based on studies of Kim et al. [38] and Farrukh et al. [32]. Kim et al. [38] conducted a study on aquatic exercises and found cognitive benefits in a sample of 26 older adults. In a meta-analysis of various exercise types, Farrukh et al. [32] reported an effect size of 0.53 on the benefits of exercise on cognitive functions. Using these studies as references, we utilized G*Power software version 3.1.9.4 (Heinrich-Heine-Universität Düsseldorf, Germany) with an effect size of 0.53, an α error probability of 0.05, and a power (1-β error) of 0.90. The analysis indicated a need for a minimum of 30 participants (15 in each group).

In this study, a single evaluator assessed all participants. The evaluator was a psychologist with previous experience administering the tests used in this study. With regard to randomization, we initially proposed using randomly selected block sizes to ensure equal distribution of participants between the experimental and control groups [39]. However, this trial was performed during the COVID-19 pandemic, and because of the recommendation for social isolation at certain points in time, some participants expressed concerns about attending in-person sessions. To prevent sample losses, we faced allocation limitations and decided to allocate participants by convenience. Participants who felt insecure about attending in-person aquatic exercises were assigned to the control group. We acknowledge the bias associated with non-randomization of the participants. However, this approach was necessary to control for type I (α) and type II (β) statistical errors. 

### 2.2. Outcomes

The main outcome measures were cognitive functions, which were assessed at baseline and re-evaluated after 12 weeks. Evaluations were conducted in a controlled environment in terms of space, lighting, temperature, and ambient noise. The order of the tests was random to avoid potential bias. 

Cognitive functions were analyzed using Raven’s Progressive Matrices [40] test and the Wisconsin Card Sorting Test [41]. Raven’s Progressive Matrices test is an ability test used to assess abstract reasoning. The instrument consists of a notebook containing three sets (A, AB, and B), each with 12 items. The test patterns are presented in the form of 6 × 6, 4 × 4, 3 × 3, or 2 × 2 matrices, which give the test its name. Raven’s Progressive Matrices test measures a subject’s speed and accuracy in interpreting information and identifying relationships between shapes and patterns. In this test, a higher number of correct responses indicates a better cognitive performance.

The Wisconsin Card Sorting Test is a problem-solving instrument consisting of a set of 128 cards with three distinct characteristics: color, shape, and number of figures. In the test, participants learn the sorting rule through feedback provided after each response. After a certain number of correct matches, the rule changes, requiring the participant to shift to a new mode of classification. In this study, the test parameters included the number of trials, number of correct answers, number of perseverative answers, number of errors, number of perseverative errors (errors made after a rule change), and number of non-perseverative errors. Better performance on this test is indicated by a higher number of correct answers, lower number of trials needed to finish the test, and lower number of perseverative answers and errors.

### 2.3. Therapeutic Protocol

The aquatic exercise protocol was exclusively implemented in the experimental group. The control group was instructed to maintain their basic activities without any change. The aquatic exercise program was designed with the goal of improving specific aspects of cognitive functions, including attention, concentration, memory, and executive processing. The intervention consisted of group classes held twice a week for 12 weeks. The group classes were led by a licensed physical therapist, supported by research assistants. Water float dumbbells, weights, noodles, and exercise discs were used in all the sessions.

The classes were held in indoor poll of the outpatient clinic, with a water depth of 1.2 m and temperature of 30–32 °C. Each class lasted 50 min, including 10 min of warm-up exercises, 30 min of cognitive–motor training, and 10 min of cool-down exercises. The focus of the intervention was integrating motor and cognitive challenges. The initial part of the program was applied at a low intensity to help the participants adapt to the water temperature. The main part of the exercise consisted of a continuous aerobic program that incorporated resistance training and muscle activation exercises. During this phase, participants engaged in walking activities as well as exercises for the upper and lower limbs at a moderate to challenging level of intensity. Cognitive components were included, such as naming animals, fruits, days of the week, or months; counting backward; and defining figures and colors. The final part of the aquatic exercise program focused on returning to a calm state through relaxing exercises. 

The circuit activities were changed every 2 weeks while maintaining the goals of stimulation. Progression was achieved by increasing the speed of exercise, number of repetitions, and range of motion of the movements. The workload of the exercises was evaluated using the Borg scale [42]. 

### 2.4. Statistical Analysis

Statistical analysis was performed in several steps. First, we characterized the variables using mean and standard deviation (SD). Second, we assessed the parametric assumptions of the data. For data that did not meet the assumptions of normality, homogeneity of variance, and sphericity, we applied a logarithmic transformation. Third, using independent Student’s *t*-tests, we compared the baseline characteristics of the groups for Raven’s Progressive Matrices test and the Wisconsin Card Sorting Test. Finally, multiple and univariate analyses of variance with Wilks’ λ test were performed to examine the main effect of group (experimental × control), time (baseline × after 12 weeks), and interactions (group × time). Effect sizes (*η*^2^*p*) were used to accurately assess the impact of the aquatic exercise program. Effect sizes were reported only when a level of significance was achieved (*p* < 0.05).

## 3. Results

Thirty-six participants were originally recruited. Five individuals refused to participate due to problems caused by the COVID-19 pandemic and difficulties in attending the outpatient clinic. Thirty-one participants completed the study (16 in the experimental group and 15 in the control group). Table 1 shows the baseline characteristics of the participants. The groups were comparable in terms of sample size, age, score on the Mini-Mental State Examination test, Raven’s Progressive Matrices test, and Wisconsin Card Sorting Test.

### 3.1. Raven’s Progressive Matrices

Both the pre- and post-assessment data of Raven’s Progressive Matrices test confirmed the assumptions of normality and homogeneity of variance. Consequently, a logarithmic conversion of these data was not necessary. A repeated-measures analysis of variance revealed that the aquatic exercise had a beneficial effect on the cognitive functions of individuals in the experimental group compared with those in the control group (*p* = 0.046, *η*^2^*p* = 0.131). That is, while the participants in the control group remained stable, individuals in the experimental group showed an improvement in terms of their cognitive scores. Table 2 shows the groups’ initial and final scores on Raven’s Progressive Matrices test.

### 3.2. Wisconsin Card Sorting Test

All variables in the Wisconsin Card Sorting Test were transformed logarithmically to meet the assumptions of normality, homogeneity of variance, and sphericity. After this adjustment, a multivariate analysis revealed significant benefits of the aquatic exercise on cognitive functions when comparing the experimental and control groups (*p* = 0.001, *η*^2^*p* = 0.589). A univariate analysis indicated that the benefits of the aquatic exercise were observed in terms of the number of trials (*p* = 0.001, *η*^2^*p* = 0.478), number of errors (*p* = 0.001, *η*^2^*p* = 0.458), and number of non-perseverative errors (*p* = 0.001, *η*^2^*p* = 0.302). Both groups exhibited similar declines in the number of correct answers after 12 weeks (*p* = 0.034, *η*^2^*p* = 0.146). No significant effect of the aquatic exercise program was observed on the number of perseverative answers (*p* = 0.084) and on the number of perseverative errors (*p* = 0.234). However, for perseverative errors, the experimental group showed a positive oscillation, and the control group showed a negative trend (*p* = 0.008, *η*^2^*p* = 0.218). Table 3 presents their initial and final scores on the Wisconsin Card Sorting Test.

## 4. Discussion

This study investigated the benefits of a 3-month aquatic exercise program on the cognitive function of community-dwelling older adults. The findings confirmed our hypothesis that individuals who underwent aquatic training experienced improvements in specific aspects of cognitive function compared to their sedentary control counterparts. These findings are particularly relevant for both fundamental and clinical applications. 

Cognitive functions are important for carrying out daily tasks and maintaining independence in older adults [43]. There is a broad distinction between cognitive decline as a common aspect of aging and pathological conditions affecting the brain, such as mild cognitive impairment and dementia [44]. This study targeted community-dwelling older individuals without cognitive impairments, as studies addressing interventions to mitigate the cognitive decline inherent in aging are scarce.

Some people may imagine that neuroimaging exams, such as magnetic resonance imaging (MRI), positron emission tomography (PET Scan), or electroencephalography (EEG), are sufficient to assess participants’ cognition. In fact, cognitive functions do not depend on a specific brain area but rather on complex neural circuits [45]. In addition, there is a phenomenon called cognitive reserve, in which the brain compensates and finds alternate ways of completing cognitive tasks, even when its structure already shows signs of disease [46]. Neuroimaging is not an appropriate method to assess cognitive function, especially in the preclinical dementia period, when neuropsychological tests are observed to be the most robust indicators, correlated to high levels of amyloid deposition and hypoconnectivity across large-scale brain networks [47].

In this study, we used two specific neuropsychological tests to assess cognition: Raven’s Progressive Matrices test and the Wisconsin Card Sorting Test. These tests were chosen because of their ability to assess different aspects of cognitive functions and because they have been validated for older individuals, with a good level of test/retest reliability [48,49,50]. As a psychologist with previous experience in these tests assessed all of the participants, we believe that the findings reflect the cognitive profile of the individuals. The results detailed in Table 1 confirm that the cognitive scores of the participants in both the experimental and control groups were within the normal range when compared with those of healthy older adults and individuals with neurological disorders [51].

Aquatic exercise is typically used to enhance the functional capacity of older individuals and aids the physical recovery of people with disabilities [33,52,53,54]. This study aimed to investigate the benefits of aquatic exercise, particularly in terms of cognitive function. Participants in the experimental group engaged in non-automatic exercises in water, which required their attention and cognitive engagement. Since non-automatic training on land, such as performing dual tasks, has been shown to benefit older adults [21,55,56,57,58], we anticipated significant cognitive improvements resulting from the aquatic exercise program compared to the control group. The results confirmed these benefits, as observed in terms of both Raven’s Progressive Matrices test and the Wisconsin Card Sorting Test. However, as Häfele et al. [33] did not find cognitive benefits from aquatic exercise without specific cognitive training, the question remains whether aquatic exercise alone improves the cognitive function of community-dwelling older individuals.

With respect to Raven’s Progressive Matrices test, the experimental group’s scores improved from 22.5 to 25.7, whereas the control group’s scores remained stable at 20.7 and 20.8. The improvement in the experimental group was clinically and statistically significant. Since Raven’s Progressive Matrices test assesses the ability to solve complex problems through intellectual reasoning, we believe that the aquatic exercise protocol, which includes both motor and cognitive challenges, stimulated the cognitive functions of the participants. We encourage new studies to compare the effects of motor-cognitive exercises in water with those of pure motor exercises to determine whether the benefits are comparable.

The Wisconsin Card Sorting Test was analyzed using both multivariate and univariate tests. Multivariate analysis is important for instruments comprising several variables because it evaluates the pattern of all variables collectively and reduces the type 1 (α) statistical error [59]. By including all the variables in the statistical model, we identified the benefits of water exercise on cognitive functions in the experimental group. The univariate analyses, however, showed that the benefit was largely seen in terms of the number of trials, errors, and non-perseverative errors. At the final assessment, the participants in the experimental group required fewer trials to perform the test and made fewer errors (both total and non-perseverative errors, which refer to a qualitative change in the search for the right strategy) than the control group. This result is important because it shows that aquatic exercise activates cognitive areas related to problem-solving tasks.

However, not all variables of the Wisconsin Card Sorting Test improved with the aquatic exercise protocol. The experimental and control groups presented similar results in terms of the number of correct answers and the number of perseverative answers. The experimental group showed a slight improvement in the number of perseverative errors, whereas the control group showed a decline, although the group × time interaction was not statistically significant. We attribute the lack of improvement in these variables to two specific factors. First, it is possible that aquatic exercise does not improve all cognitive and mental components, as shown by Jackson et al. [60] and Häfele et al. [33]. Second, the aquatic exercise protocol used in this study, which consisted of sessions held twice a week for 3 months, may not have been sufficient to improve all cognitive components in the community-dwelling older individuals. Incorporating more sessions could yield different outcomes. 

We emphasize that an important aspect of this study is the incorporation of cognitive challenges alongside aquatic exercises. Previous studies have identified the benefits of simultaneously performing cognitive and motor tasks on land [61,62,63,64]. However, studies examining cognitive–motor training in water are scarce. Our study design demonstrated the benefits of combining motor and cognitive tasks in an aquatic setting. Nevertheless, it does not allow us to conclude whether aquatic exercise alone can improve cognitive functions to the same extent as when combined with cognitive components.

Finally, Table 2 and Table 3 show that the effect sizes of the aquatic exercise varied across the instruments. While the effect size for Raven’s Progressive Matrices test was 0.131, that for the Wisconsin Card Sorting Test varied from 0.146 to 0.478. Readers should understand that effect sizes allow researchers to make judgments about the relevance of findings [65]. On the one hand, one might interpret that the benefits of aquatic exercises were greater on the Wisconsin Card Sorting Test than on Raven’s Progressive Matrices test. However, on the other hand, one should not focus solely on the data. The improvement on Raven’s Progressive Matrices test may be as clinically relevant as the improvements seen in the number of trials, number of errors, and number of non-perseverative errors on the Wisconsin Card Sorting Test, even with a smaller effect size. Thus, while effect sizes are important and should be interpreted alongside the significance level, they should not be restricted to numbers alone.

### Limitations

Our findings should be interpreted in light of the following limitations. First, the results were restricted to community-dwelling older individuals. Second, the control group was not randomly allocated. This introduces the possibility of a hidden effect that was not measured in this study. Third, this study was conducted over a relatively short period, making it challenging to observe the full extent of the changes resulting from the aquatic exercise. Fourth, this study did not account for certain fluid mechanics aspects, such as water viscosity and refraction rate, nor did it consider some individual characteristics, such as the velocity of the participants in water. It is plausible that higher speeds could have increased the level of water resistance. We encourage further studies to carefully control for these aspects to obtain more accurate results.

## 5. Conclusions

The 3-month aquatic exercise training had positive effects on cognitive functions in community-dwelling older adults. Continued investigation is important to gain a more comprehensive understanding of the potential benefits of aquatic exercise, both with and without cognitive training, in preventing cognitive decline associated with aging.

## Figures and Tables

**Table 1 brainsci-14-00703-t001:** Baseline characteristics of the participants.

Variables	Groups		*t_value_*	*p*
	Experimental	Control		
Sample size, n	16	15	-	0.857
Age, n	67.2 (4.4)	69.1 (6.4)	−0.952	0.349
Mini-Mental State Examination, pts	26.4 (2.5)	27.0 (2.7)	−0.590	0.560
Raven’s Progressive Matrices, pts	22.5 (8.0)	20.7 (8.5)	0.895	0.557
Number of trials, n	113.6 (12.5)	112.2 (20.1)	0.408	0.687
Number of correct answers, n	76.1 (6.4)	73.1 (7.9)	1.212	0.235
Number of perseverative answers, n	32.4 (9.8)	35.7 (12.8)	−0.821	0.418
Number of errors, n	37.5 (14.8)	38.7 (19.4)	0.306	0.762
Number of perseverative errors, n	18.9 (7.2)	22.3 (10.9)	−0.994	0.330
Number of non-perseverative errors, n	17.7 (11.0)	16.1 (9.3)	0.654	0.518

**Table 2 brainsci-14-00703-t002:** Initial and final assessment on Raven’s Progressive Matrices test.

Variable	Groups	Assessment		ANOVA Main Effect		
		Initial	Final	Group	Time	Interaction
Raven’s Progressive Matrices, pts	Experimental	22.5 (8.0)	25.7 (5.1)	*F_value_* = 1.904 *p* = 0.178	*F_value_* = 4.742 *p* = 0.038 *η*^2^*p* = 0.141	*F_value_* = 4.361 *p* = 0.046 *η*^2^*p* = 0.131
	Control	20.7 (8.5)	20.8 (5.9)			

**Table 3 brainsci-14-00703-t003:** Initial and final assessment of the participants on the Wisconsin Card Sorting Test.

Variable	Groups	Assessment		ANOVA Main Effect		
		Initial	Final	Group	Time	Interaction
Number of trials, n	Experimental	113.6 (12.5)	102.5 (12.1)	*F_value_* = 1.932 *p* = 0.175	*F_value_* = 0.458 *p* = 0.504	*F_value_* = 26.530 *p* = 0.001 *η*^2^*p* = 0.478
	Control	112.2 (20.1)	120.3 (14.4)			
Number of correct answers, n	Experimental	76.1 (6.4)	73.6 (3.7)	*F_value_* = 6.454 *p* = 0.170	*F_value_* = 4.957 *p* = 0.034 *η*^2^*p* = 0.146	*F_value_* = 1.108 *p* = 0.301
	Control	73.1 (7.9)	67.3 (10.7)			
Number of perseverative answers, n	Experimental	32.4 (9.8)	31.4 (7.7)	*F_value_* = 3.001 *p* = 0.094	*F_value_* = 2.587 *p* = 0.119	*F_value_* = 3.202 *p* = 0.084
	Control	35.7 (12.8)	48.8 (11.3)			
Number of errors, n	Experimental	37.5 (14.8)	29.0 (10.9)	*F_value_* = 2.996 *p* = 0.094	*F_value_* = 1.447 *p* = 0.239	*F_value_* = 24.511 *p* = 0.001 *η*^2^*p* = 0.458
	Control	38.7 (19.4)	52.9 (18.8)			
Number of perseverative errors, n	Experimental	18.9 (7.2)	16.7 (5.9)	*F_value_* = 0.042 *p* = 0.839	*F_value_* = 8.102 *p* = 0.008 *η*^2^*p* = 0.218	*F_value_* = 1.478 *p* = 0.234
	Control	22.2 (10.9)	29.1 (9.9)			
Number of non-perseverative errors, n	Experimental	17.7 (11.0)	12.3 (6.6)	*F_value_* = 1.164 *p* = 0.290	*F_value_* = 0.1001 *p* = 0.754	*F_value_* = 12.519 *p* = 0.001 *η*^2^*p* = 0.302
	Control	6.1 (9.3)	22.1 (9.8)			

## Data Availability

The original contributions presented in the study are included in the article and Appendix A; further inquiries can be directed to the corresponding author.

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
