# Peer review of "A Controlled Clinical Trial on the Effects of Aquatic Exercise on Cognitive Functions in Community-Dwelling Older Adults"

_brainsci, 2024, doi:10.3390/brainsci14070703_

Round 1

Reviewer 1 Report

Comments and Suggestions for Authors

The manuscript “A Controlled Clinical Trial on the Effects of Aquatic Exercise on Cognitive Functions in Community-Dwelling Older Adults” by Renata Terra de Oliveira is a research article. The authors found that compared with the control group, participants who underwent aquatic exercises showed positive outcomes in the Raven's Progressive Matrices and Wisconsin Card Sorting Test.  Complementary analyses on the Wisconsin Card Sorting Test revealed that the benefits of the aquatic exercise were observed in the number of trials, number of errors, and number of non-perseverative errors. Therefore, the authors concluded that three months of aquatic exercise was beneficial for stimulating specific aspects of cognitive functions in community-dwelling older individuals. In general, this research article is critical in this field and contains essential contents. However, I have several comments before this manuscript is accepted for publication.

1. The statistical results in the abstract are not needed in the abstract in my opinion.

2. It would be better to describe the number of men and female among 61 subjects.

3. The statistical assessment was performed using student’s t-test and ANOVA. Please add t and f vales in the text.

4. This study investigated the benefits of a 3-month aquatic exercise program on cognitive functions in community-dwelling older adults. How do the authors think about the period of aquatic exercise program? Is the 3-month aquatic exercise program most suitable for improving cognitive functions?

Author Response

RESPONSE TO REVIEWERS

Manuscript ID: behavsci-3101965

Manuscript Title: A Controlled Clinical Trial on the Effects of Aquatic Exercise on Cognitive Functions in Community-Dwelling Older Adults

We would like to express our appreciation for the time and effort put in by the editor and the reviewers. Their comments and suggestions have improved the quality of the manuscript.

REVIEWER #1

Comment 1: The manuscript “A Controlled Clinical Trial on the Effects of Aquatic Exercise on Cognitive Functions in Community-Dwelling Older Adults” by Renata Terra de Oliveira is a research article. The authors found that compared with the control group, participants who underwent aquatic exercises showed positive outcomes in the Raven's Progressive Matrices and Wisconsin Card Sorting Test. Complementary analyses on the Wisconsin Card Sorting Test revealed that the benefits of the aquatic exercise were observed in the number of trials, number of errors, and number of non-perseverative errors. Therefore, the authors concluded that three months of aquatic exercise was beneficial for stimulating specific aspects of cognitive functions in community-dwelling older individuals. In general, this research article is critical in this field and contains essential contents. However, I have several comments before this manuscript is accepted for publication.

Response 1: We thank for the positive feedback and for the valuable suggestions, which have enhanced the quality of the study.

Comment 2: The statistical results in the abstract are not needed in the abstract in my opinion.

Response 2: The inclusion of statistical results in the abstract section is a topic of debate among researchers. However, as the abstract offers an overview of the study and we have reported effect sizes (which are often omitted in many studies), we believe that retaining this data enhances the attractiveness of our study. Therefore, we have chosen to keep the statistical results.

Comment 3: - It would be better to describe the number of men and female among 61 subjects.

Response 3: We included that information in the results section. Now it reads: “The sample consisted of 31 individuals, 17 women and 14 men, mean age of 68.1 yrs (SD, 5.5), who were allocated to either the experimental (intervention) or control (no intervention) group.

Comment 4: The statistical assessment was performed using student’s t-test and ANOVA. Please add t and F vales in the text.

Response 4: Thank you for bringing this to our attention. We included tvalues and Fvalues on tables 1, 2, and 3.

Comment 5: This study investigated the benefits of a 3-month aquatic exercise program on cognitive functions in community-dwelling older adults. How do the authors think about the period of aquatic exercise program? Is the 3-month aquatic exercise program most suitable for improving cognitive functions?

Response 5: In the discussion section, we address factors that may explain why not all variables of the Wisconsin Card Sorting Test benefited from the intervention, including the 3-month duration. Despite this, the 3-month period yielded significant results, as detailed in Tables 2 and 3. When asked if we believe a 3-month aquatic exercise program is suitable for improving cognitive functions, our answer is yes. However, further studies are needed to investigate whether a longer period may produce even better results. We have enhanced the discussion on this topic, as well as the conclusion section.

Editor

The revision of the manuscript was performed in accordance with the comments/concerns raised by the reviewers. All the suggestions made by the reviewers were followed or clarified. The revised parts are highlighted in red in the revised manuscript to highlight the changes. We appreciate the comments that indeed helped to improve the quality of the manuscript.

Reviewer 2 Report

Comments and Suggestions for Authors

The article aims to investigate the effect of aquatic exercise on cognitive functions in community-dwelling older adults. The aim of the study is to investigate the potential benefits of a 3-month aquatic exercise program on cognitive functions in community-dwelling older adults. The authors formulate a clear research hypothesis and work to test it.

The topic that is developed in this article is very important and relevant from both fundamental and applied points of view. The gap of the study is the lack of clear knowledge about the potential benefits of aquatic exercises on cognitive functions.

Compared with other published material, this study adds to the subject area the information about how concurrent aquatic exercises would optimize brain plasticity and mitigate age-related decline.

I have no questions about the methodology of the study. Of course, the study sample is small, but the authors justified the use of such a sample size.

The references are appropriate.

I would also like to note that the article is well structured, written in a literate language, and represents a valuable and accessible scientific study.

I would like to see in Discussion the citation of the sources that the authors use in the introduction. At the moment the cited sources are different for the introduction and for Discussion.

Author Response

RESPONSE TO REVIEWERS

Manuscript ID: behavsci-3101965

Manuscript Title: A Controlled Clinical Trial on the Effects of Aquatic Exercise on Cognitive Functions in Community-Dwelling Older Adults

We would like to express our appreciation for the time and effort put in by the editor and the reviewers. Their comments and suggestions have improved the quality of the manuscript.

REVIEWER #2

Comment 1: The article aims to investigate the effect of aquatic exercise on cognitive functions in community-dwelling older adults. The aim of the study is to investigate the potential benefits of a 3-month aquatic exercise program on cognitive functions in community-dwelling older adults. The authors formulate a clear research hypothesis and work to test it.

Response 1: We thank for the positive feedback and for the valuable suggestions, which have enhanced the quality of the study.

Comment 2: The topic that is developed in this article is very important and relevant from both fundamental and applied points of view. The gap of the study is the lack of clear knowledge about the potential benefits of aquatic exercises on cognitive functions.

Response 2: In this study, we aimed to enhance the discussion about the potential benefits of aquatic exercise on cognitive functions. The current body of research is limited, likely due to the small number of studies addressing this topic. Upon encountering the special issue “Effects of Cognitive Training on Executive Function and Cognition” presented in Brain Sciences, we realized that our study aligns well with its scope. Therefore, we submitted our study to Brain Sciences for a thorough review. Thank you for the positive feedback and valuable suggestions.

Comment 3: Compared with other published material, this study adds to the subject area the information about how concurrent aquatic exercises would optimize brain plasticity and mitigate age-related decline.

Response 3: We thank for the positive feedback.

Comment 4: I have no questions about the methodology of the study. Of course, the study sample is small, but the authors justified the use of such a sample size.

Response 4: We thank for the feedback. We acknowledge that the sample size is limited. However, the calculated sample size justifies the number of subjects in this study, controlling for type 1 and type 2 statistical errors. This means that, although the number of individuals in each group is low, it is statistically sufficient to confirm the findings. We have discussed this in the methods section and explained its implications.

Comment 5: The references are appropriate.

Response 5: We thank for the feedback.

Comment 6: I would also like to note that the article is well structured, written in a literate language, and represents a valuable and accessible scientific study.

Response 6: We thank for the feedback. We conducted this study according to the CONSORT guidelines and registered it with the Brazilian Clinical Trials Registry. We believe that this rigorous approach was essential for making the study understandable and replicable.

Comment 7: I would like to see in Discussion the citation of the sources that the authors use in the introduction. At the moment the cited sources are different for the introduction and for Discussion.

Response 7: We agree that the discussion section needed improvement on this topic. We included five new references and added more text to enhance the comparison of our study with previously published work. Please check the discussion section to see the improvements highlighted in red.

Editor

The revision of the manuscript was performed in accordance with the comments/concerns raised by the reviewers. All the suggestions made by the reviewers were followed or clarified. The revised parts are highlighted in red in the revised manuscript to highlight the changes. We appreciate the comments that indeed helped to improve the quality of the manuscript.

Reviewer 3 Report

Comments and Suggestions for Authors

In the manuscript entitled “A Controlled Clinical Trial on the Effects of Aquatic Exercise on Cognitive Functions in Community-Dwelling Older Adults”, the authors presented a controlled clinical trial conducted in a single site in Brazil. Using the Raven's Progressive Matrices and Wisconsin Card Sorting Test, the 3-month intervention, administered twice weekly, significantly improved cognitive functions. Compared to controls, the exercise group displayed significant improvements in cognitive performance metrics, with notable effect sizes. From these observations, the authors concluded that aquatic exercise enhanced cognitive functions in community-dwelling older individuals. The current findings are interesting and the manuscript is clearly written.

Comments: 

1) To make it clearer for readers, the authors are advised to elaborate in the introduction section on the novelty of the present work. Please, elaborate on the sharp differences that highlight how the study is different from previous literature that has already addressed the effect of aquatic exercises on cognitive function in elderly human subjects. An example is the work of Häfele et al., 2023 (Water-Based Training Programs Improve Functional Capacity, Cognitive and Hemodynamic Outcomes? The ACTIVE Randomized Clinical Trial, Res Q Exerc Sport. 2023 Mar;94(1):24-34, PMID: 35294330). What is already described by the authors regarding this issue needs to be further elaborated.

2) The title of the present work needs to be modified to reflect the main finding of the current work. Since the current study presents outcomes extracted from in Brazil, this fact should be clearly indicated in the title. 

3) A major issue is that the current study lacks a description/stratification of the cases based on the severity of cognitive decline symptoms. This would have been a better approach for making the best use of the current findings.

4) In the current study, the cognitive function was examined by Raven's progressive matrices and Wisconsin card sorting test, without providing common medical scans to characterize dementia, such as EEG, MRI, or PET Scan. In fact, a combination of these tests would substantiate the outcomes and conclusions from the present work.

5) In the Material and Methods section, to make it clearer for readers, the authors are advised to provide a rationale for choosing the Raven's Progressive Matrices and Wisconsin Card Sorting Test, and how they are more convenient than other tests that examine cognitive function in human subjects.

6) A major point in the present work is the limited small sample size (n = 31) which may affect the generalizability of the present findings.

7) Caution should be applied to the conclusions of the current study since the data extracted was based on a single site in Brazil. Thus, conclusions about other sites in Brazil, or worldwide cannot be extracted. This point needs to be clarified in the discussion section.

8) The discussion contains interesting points. However, the authors should put more effort into integrating/replacing their findings in the context of current knowledge. From a mechanistic perspective, the authors are advised to elaborate on how specific the physical activity involved in aquatic exercises enhances cognitive functions compared to other forms of exercise.

9) In the discussion section, the authors are advised to comment on the differences observed between the substantial (η²p = 0.589) effect size for the Wisconsin card sorting test, compared to the moderate (η²p = 0.131) effect size for Raven's progressive matrices. What are the potential implications?  

10) to enrich, the discussion, the authors are advised to comment on the apparent controversy between the current findings revealing enhanced cognitive ability in community-dwelling older individuals, compared to the outcomes of the study by Häfele et al., 2023 (PMID: 35294330) that revealed no significant enhancement of the cognitive functions in older women in response to aquatic exercises.   

Author Response

RESPONSE TO REVIEWERS

Manuscript ID: behavsci-3101965

Manuscript Title: A Controlled Clinical Trial on the Effects of Aquatic Exercise on Cognitive Functions in Community-Dwelling Older Adults

We would like to express our appreciation for the time and effort put in by the editor and the reviewers. Their comments and suggestions have improved the quality of the manuscript.

REVIEWER #3

Comment 1: In the manuscript entitled “A Controlled Clinical Trial on the Effects of Aquatic Exercise on Cognitive Functions in Community-Dwelling Older Adults”, the authors presented a controlled clinical trial conducted in a single site in Brazil. Using the Raven's Progressive Matrices and Wisconsin Card Sorting Test, the 3-month intervention, administered twice weekly, significantly improved cognitive functions. Compared to controls, the exercise group displayed significant improvements in cognitive performance metrics, with notable effect sizes. From these observations, the authors concluded that aquatic exercise enhanced cognitive functions in community-dwelling older individuals. The current findings are interesting and the manuscript is clearly written.

Response 1: We thank for the feedback and for the valuable suggestions, which have enhanced the quality of the study.

Comment 2: To make it clearer for readers, the authors are advised to elaborate in the introduction section on the novelty of the present work. Please, elaborate on the sharp differences that highlight how the study is different from previous literature that has already addressed the effect of aquatic exercises on cognitive function in elderly human subjects. An example is the work of Häfele et al., 2023 (Water-Based Training Programs Improve Functional Capacity, Cognitive and Hemodynamic Outcomes? The ACTIVE Randomized Clinical Trial, Res Q Exerc Sport. 2023 Mar;94(1):24-34, PMID: 35294330). What is already described by the authors regarding this issue needs to be further elaborated.

Response 2: Thank you for bringing this to our attention. We included the following text to addres this suggestion: “One of the few studies addressing the impact of aquatic exercise on cognitive function was conducted by Häfele et al. [33]. The researchers assessed 52 participants, with 35 who underwent water-based aerobic training and 17 remained sedentary in the control group. The findings showed no benefits of the intervention on the cognitive scores of the Mini-Mental State Examination. This raises the question of whether aquatic exercise alone can enhance cognitive function in older individuals compared with a combination of wa-ter and cognitive training. By contrast, Campbell et al. [34] used a different approach. The researchers enrolled 31 individuals in a water exercise program combined with cognitive training, whereas the control group of 36 participants underwent land-based exercise with cognitive training. Notably, all participants in this study had mild cognitive impairment, unlike in Häfele et al. [33]. The authors found that individuals in the aquatic-based exercise plus cognitive training group experienced significant cognitive benefits, whereas those in the land-based exercise plus cognitive training group showed limited improvement. These results suggest that aquatic exercise combined with cognitive training can improve cognition in individuals with mild cognitive impairment. However, it remains unclear whether water exercise improves cognition in cognitively preserved older adults”.

Comment 3: The title of the present work needs to be modified to reflect the main finding of the current work. Since the current study presents outcomes extracted from in Brazil, this fact should be clearly indicated in the title.

Response 3: We appreciate the suggestion. However, we believe that restricting the results to a Brazilian sample would make the study less attractive to readers. While our sample consists exclusively of Brazilian participants, the tests used (Raven's Progressive Matrices and the Wisconsin Card Sorting Test) are validated across multiple countries and languages. Therefore, we believe our findings are applicable to other populations as well. We respectfully disagree with the reviewer and we have retained the original title.

Comment 4: A major issue is that the current study lacks a description/stratification of the cases based on the severity of cognitive decline symptoms. This would have been a better approach for making the best use of the current findings.

Response 4: None of the participants exhibited symptoms of cognitive decline. We included this information in the text as follows: “All the participants were screened for dementia. Individuals who exhibited scores on the Mini-Mental State Examination lower than the cutoff values adjusted for education and age established for the local population were excluded from the study [36,37].

Comment 5: In the current study, the cognitive function was examined by Raven's progressive matrices and Wisconsin card sorting test, without providing common medical scans to characterize dementia, such as EEG, MRI, or PET Scan. In fact, a combination of these tests would substantiate the outcomes and conclusions from the present work.

Response 5: In the discussion section, we highlighted that EEG, MRI, or PET scans are not the most suitable instruments for assessing cognitive functions. These neuroimaging methods provide valuable insights into brain structure and activity but do not directly measure cognitive abilities. Differently, instruments like the Raven's Progressive Matrices and the Wisconsin Card Sorting Test are more appropriate as they directly assess cognitive functions relevant to our investigation. Please see the highlighted text in the discussion section for further details. We copied it here: “Some people may imagine that using neuroimaging exams, such as magnetic resonance imaging (MRI), positron emission tomography (PET Scan), or electroencephalography (EEG), is sufficient to assess participants’ cognition. In fact, cognitive functions do not depend on a specific brain area but rather on complex neural circuits [46]. In addition, there is a phenomenon called cognitive reserve, in which the brain compensates and finds alternate ways of completing cognitive tasks, even when its structure already shows signs of disease [47]. Neuroimaging is not an appropriate method to assess cognitive functions, especially in the preclinical dementia period, where neuropsychological tests were ob-served to be the most robust indicators, correlating with high levels of amyloid deposition and hypoconnectivity across large-scale brain networks [48]”.

Comment 6: In the Material and Methods section, to make it clearer for readers, the authors are advised to provide a rationale for choosing the Raven's Progressive Matrices and Wisconsin Card Sorting Test, and how they are more convenient than other tests that examine cognitive function in human subjects.

Response 6: We have included that information in the discussion section. Please see the highlighted text in the discussion section for further details. We copied it here: “In this study, we used two specific neuropsychological tests to assess cognition: Raven's Progressive Matrices and Wisconsin Card Sorting Test. These tests were chosen because of their ability to assess different aspects of cognitive functions and their valida-tion for older individuals, with good test-retest reliability [49-51]. As a psychologist with previous experience in these tests assessed all participants, we believe that the findings reflect the cognitive profile of the individuals”.

Comment 7: A major point in the present work is the limited small sample size (n = 31) which may affect the generalizability of the present findings.

Response 7: We acknowledge that the sample size is limited. However, the calculated sample size justifies the number of subjects in this study, controlling for type 1 and type 2 statistical errors. This means that, although the number of individuals in each group is relative low, it is statistically sufficient to confirm the findings. We have discussed this in the methods section and explained its implications.

Comment 8: Caution should be applied to the conclusions of the current study since the data extracted was based on a single site in Brazil. Thus, conclusions about other sites in Brazil, or worldwide cannot be extracted. This point needs to be clarified in the discussion section.

Response 8: Please refer to our response to comment 3, where we explain why it is unnecessary to restrict the results to Brazil. Following the rationale of this discussion, should all studies conducted in a single site state the location in the conclusion? I agree that if we explored a disease unique to Brazil, this would be pertinent. However, this was not the case.

Comment 9: The discussion contains interesting points. However, the authors should put more effort into integrating/replacing their findings in the context of current knowledge. From a mechanistic perspective, the authors are advised to elaborate on how specific the physical activity involved in aquatic exercises enhances cognitive functions compared to other forms of exercise.

Response 9: Thank you for the suggestion on improving this discussion. We included the following text to address this concern: “Aquatic exercise is typically used to enhance the functional capacity of older individuals and aid in the physical recovery of people with disabilities [53-56]. This study aimed to investigate the additional benefits of aquatic exercise, particularly in terms of cognitive functions. Participants in the experimental group  engaged in non-automatic exercises in water, which required attention and cognitive engagement. Since non-automatic training on land, such as dual tasks, has been shown to benefit older adults [21, 57-60], we anticipated significant cognitive improvements resulting from aquatic exercise compared to the control group. The results confirmed these benefits, as observed in both the Raven's Progressive Matrices and the Wisconsin Card Sorting Test. However, as Häfele et al. [33] did not find cognitive benefits from aquatic exercise without a specific cognitive training, the question remains whether aquatic exercise alone improves cognitive functions in community-dwelling older individuals”.

Comment 10: In the discussion section, the authors are advised to comment on the differences observed between the substantial (η²p = 0.589) effect size for the Wisconsin card sorting test, compared to the moderate (η²p = 0.131) effect size for Raven's progressive matrices. What are the potential implications?

Response 10: included the following text to address this concern: “Finally, Tables 2 and 3 show that the effect sizes of aquatic exercises varied across the instruments. While the effect size for Raven's Progressive Matrices was 0.131, that for the Wisconsin Card Sorting Test varied from 0.146 to 0.478. Readers should understand that effect sizes allow researchers to make judgments about the relevance of findings [67]. On the one hand, one might interpret that the benefits of aquatic exercises were greater on the Wisconsin Card Sorting Test than on Raven's Progressive Matrices. However, on the other hand, one should not focus solely on the data. The improvement on Raven's Progressive Matrices may be clinically relevant as the benefits seen in the number of trials, number of errors, and number of non-perseverative errors on the Wisconsin Card Sorting Test, even with a smaller effect size. Thus, while effect sizes are important and should be interpreted alongside the significance level, they should not be restricted to numbers alone”.

Comment 11: to enrich, the discussion, the authors are advised to comment on the apparent controversy between the current findings revealing enhanced cognitive ability in community-dwelling older individuals, compared to the outcomes of the study by Häfele et al., 2023 (PMID: 35294330) that revealed no significant enhancement of the cognitive functions in older women in response to aquatic exercises.

Response 11: We agree that the discussion section needed improvement on this topic. We included five new references and added more text to enhance the comparison of our study with previously published work. Please check the discussion section to see the improvements highlighted in red.

Editor

The revision of the manuscript was performed in accordance with the comments/concerns raised by the reviewers. All the suggestions made by the reviewers were followed or clarified. The revised parts are highlighted in red in the revised manuscript to highlight the changes. We appreciate the comments that indeed helped to improve the quality of the manuscript.

Round 2

Reviewer 3 Report

Comments and Suggestions for Authors

The authors have adequately addressed the raised comments.